# Resolving active species during the carbon monoxide oxidation over Pt(111) on the microsecond timescale

Calley N. Eads[1], Weijia Wang [1], Ulrike Küst [2,3], Julia Prumbs [2], Robert H. Temperton [1], Mattia Scardamaglia [1], Joachim Schnadt[1,2,3], Jan Knudsen[1,2,3] & Andrey Shavorskiy [1] ✉

Catalytic studies traditionally rely on steady-state conditions resulting in time-averaged datasets that do not differentiate between active and spectator species. This limitation can cause misinterpretations of catalytic function, as the signal of short-lived intermediates responsible for producing desired reaction products is often masked by more intense spectator species. Time-resolved ambient pressure X-ray photoelectron spectroscopy (tr-APXPS) mitigates this issue by combining microsecond time resolution under reaction conditions. Using tr-APXPS, we investigate the oxidation of CO over Pt(111) by concurrently tracking reaction products, surface intermediates, and catalyst response. Our findings reveal that chemisorbed oxygen, rather than Pt surface oxide, is the main species reacting with CO to form $CO_2$, supporting a primary Langmuir-Hinshelwood mechanism. The results shed new light on a heavily-debated reaction in catalysis. Beyond using CO pulses to determine active species, we demonstrate how careful tuning of pulsing parameters can be used for dynamic catalyst operation to enhance $CO_2$ formation.

The development of novel experimental methods enabling the study of materials and processes with higher complexity than before can significantly impact even well-established research areas, sometimes prompting a rethinking of existing theories. A classic example is the investigation of carbon monoxide oxidation over platinum-group metals. In the 1980s and 1990s, surface science experiments clearly supported the Langmuir-Hinshelwood (LH) mechanism, identifying chemisorbed oxygen as the sole reactive species[1–3]. However, with the advent of operando techniques in the 2000s, the study of catalytic reactions under (near) ambient pressures and technologically relevant temperatures provided evidence of platinum oxide formation, which sometimes correlated with enhanced catalytic activity[4–6]. These new operando findings led to a revision of the existing understanding of the CO oxidation mechanism, suggesting that platinum oxide is the catalytically most active species and that the oxidation proceeds through the Mars-van-Krevelen (MvK) mechanism[7,8]. However, other operando studies have found no evidence of platinum oxide formation[9–11] or suggested that platinum oxide plays only a spectator role[12–15], supporting the original LH mechanism with metallic platinum as the active site[16,17]. As it appears today, the mechanism of CO oxidation over platinum under (near) ambient conditions remains unresolved. Settling this debate requires developing advanced experimental capabilities that can explore new dimensions of the reaction, such as spatial or time resolution, in combination with operando capabilities[6,18,19]. We propose a time-resolved approach to differentiate active from spectator species in CO oxidation of Pt.

Most often, operando catalyst research is conducted under steady-state conditions, yielding time-averaged data that overlook the dynamic behavior of reaction species[18,20–22]. This approach can result in observational claims about the slowest reaction step[23] and the most abundant intermediates, which are not necessarily the active species[24,25]. To better understand the catalytic mechanism, it is

[1]MAX IV Laboratory, Lund University, Lund, Sweden. [2]Division of Synchrotron Radiation Research, Department of Physics, Lund University, Lund, Sweden. [3]NanoLund, Lund University, Lund, Sweden. ✉e-mail: andrey.shavorskiy@maxiv.lu.se

essential to track the exact sequence of surface transformations on a timescale comparable to that of the elementary reactions. This can be achieved by perturbing steady-state conditions with a rapid chemical potential change in the form of a gas or temperature pulse, triggering a collective response from all active sites. Time-resolved techniques then observe the synchronous evolution of surface species, effectively revealing the catalytic cycle directly[18,26,27].

The most relevant timescales for individual reaction steps are microseconds to milliseconds[18,19]. However, investigating the dynamics of surface elementary reactions on these timescales under pressure and elevated temperature conditions with high chemical specificity intact remains challenging. Leading time-resolved techniques include the temporal analysis of products (TAP)[28,29], molecular beam (MB)[30,31], steady-state isotopic transient kinetic analysis (SSITKA)[32,33], and modulation excitation spectroscopy (MES)[34,35] with phase-sensitive detection[27,36,37] or frequency space data analysis[38,39] (Supplementary Table 1). These methods use modulation parameters, such as rapid cycling of gas pressures, temperature, or concentration, and evaluate the catalytic activity based on gas phase products in mass spectrometry (MS). TAP and SSITKA offer second to sub-second resolution, while MB achieves microsecond resolution but is limited to $<10^{-3}$ mbar pressures. Nevertheless, gaining deeper insights into underlying surface processes requires coupling with spectroscopic or scattering tools, where time resolution is contingent upon the analysis technique (Supplementary Table 1). Since these methods are not typically event-averaged over many cycling events, their time resolution is limited by the signal-to-noise ratio of a single spectrum. Additionally, many MES techniques like X-ray diffraction (XRD) and X-ray absorption spectroscopy (XAS) are bulk sensitive, which limits their ability to analyze surface-active species that usually define catalytic function.

The importance of studying catalytic reactions under periodic cycling conditions extends beyond academic interest, as it has been experimentally shown to improve conversion rates in industrial CO and hydrocarbon oxidation on supported Pt and Pd catalysts[18,40–45] compared to steady-state flow regimes. For instance, automotive catalytic converters have an inherent modulation in the gas atmosphere due to a feedback loop that sets an effective air-to-fuel ratio in the exhaust gas, which constantly corrects fuel and air injections, creating rapid transient changes in gas composition. Researchers at Toyota and General Motors performed numerous experiments testing these modulation parameters on real catalysts under practical conditions and found differences in product formation[41,45]. These processes are vital for emission control and hydrogen purification, with significant economic and regulatory implications[46]. These discoveries have spurred further research into the periodic or "unsteady-state" operation of catalytic converters and other technologies, showing immense success compared to their steady-state alternatives[24,47–49]. The rate acceleration of catalytic processes induced by oscillating the catalyst between two electronic states is also the theoretical basis of surface resonance theory[50]. While redesigning existing technologies to include periodic operation is appealing[51–53], the chemical transformations driving higher conversion rates are not yet fully understood. Advancement first requires studying underlying phenomena under relevant reaction conditions using analysis techniques with adequate time resolution.

We have recently established a time-resolved ambient pressure X-ray photoelectron spectroscopy (tr-APXPS) methodology[54–56] to tackle these challenges by concurrently tracking the evolution of surface intermediates and gas phase contributions with microsecond time resolution using chemical perturbations operating at near ambient pressures and elevated temperatures[56]. This powerful method clearly separates active and spectator species by modulating conditions and directly linking them to gas phase products, probing chemical dynamics in any reversible catalytic reaction. In this work, we revolutionize the method by integrating a delay-line detector (DLD) in the

spectrometer, enabling us to continuously (i.e., throughout the entire cycle) monitor the surface and gas phase evolution with high time resolution. We stress that with the introduction of the DLD, such information can be obtained for all time delays from a single measurement. This starkly contrasts with previous measurements[56] where a single time delay could be obtained per one measurement. By introducing the DLD in this work, it was possible to increase the data collection efficiency by one to two orders of magnitude[26], allowing for a level of detail previously impossible to obtain. For example, with the new detection scheme, we could clearly identify the part of the catalytic cycle where a highly active phase transitions into a phase with low activity, pinpointing the precedence of a specific mechanism.

In this work, we studied the time evolution of reaction species in the CO oxidation of Pt(111) at 333 °C by periodically modulating CO gas into an $O_2$ stream and capturing chemical transformations with 40 µs resolution. Within this timeframe, we assess active and spectator species. We find that chemisorbed oxygen, $O_{chem}$, plays a critical role in $CO_2$ production while oxygen species in the oxide, $O_{oxide}$, mostly spectates. Our results support a primary LH mechanism with active $O_{chem}$ species and a secondary MvK mechanism involving less reactive $O_{oxide}$. Through this understanding of the structure-activity relationship, we explore how varying CO pulsing parameters can toggle $CO_2$ production on and off, enabling control of reactivity.

## Results
### Time-resolved methodology
Contrary to a typical reactor scheme that utilizes a continuous flow of all reactant gases, our tr-APXPS set-up pulses one reactant gas (CO) into a continuous stream of a different reactant gas ($O_2$) while observing time-dependent chemical composition changes via XPS using a 2D delay-line detector (DLD) (Fig. 1a). Synchronization of the gas pulses with the DLD via a function generator provides the platform to directly obtain 2D plots (maps) of kinetic energy vs. time during the duration of each gas pulse (Fig. 1b, step 1). The normal photoelectron spectrum is then reconstructed from multiple such time-resolved measurements with the kinetic energy swept between each of them (Fig. 1b, step 2). This is repeating a well-known swept mode of acquisition, the most common way of recording XPS. The key to the set-up is the aforementioned hardware synchronization, which ensures the timing structure within the 2D maps to be exactly the same allowing averaging of the time-resolved data between numerous pulses (Fig. 1b, step 3). Pertinent to our current study monitoring CO oxidation over Pt(111), Fig. 1b presents a 2D plot of C $1s$ gas phase when pulsing CO gas at 10 Hz into an $O_2$ stream averaged over 90,000 pulses taking 2.5 h. Integrating over the CO gas phase feature reveals the profile of the CO gas pulse with pulsing regions that are commonly referred to throughout the text. Briefly, before the pulse is the delay between the data acquisition start time and the CO(g) signal. The rising edge begins when the CO pulse reaches the sample and ends with CO(g) equilibration. On-pulse denotes when CO(g) is constant. The falling edge begins when CO is pumped out, reducing the CO(g) signal.

### Microsecond compositional changes during CO oxidation over Pt(111)
We employ tr-APXPS to uncover mechanistic information involving both surface and gas phase species when pulsing CO at 10 Hz (100 ms period) into a stream of $O_2$ (see Methods for experimental details). Figure 2 shows time-resolved 2D plots of XPS data of relevant core levels (top panel), and representative fits (bottom panel) at certain time points (I–IV) along the CO pulse (top left). The CO pulse profile derives from the CO(g) component of C $1s$ gas phase spectra (Fig. 2a) and is divided into four regions: (1) before pulse (<0 ms, light gray), (2) rising edge (0.0–1.6 ms, burgundy), (3) on-pulse (1.6–2.8 ms, red), and (4) falling edge (>2.8 ms, dark gray). Note that gas phase (GP) spectra were recorded 450 µm from the normal XPS position to

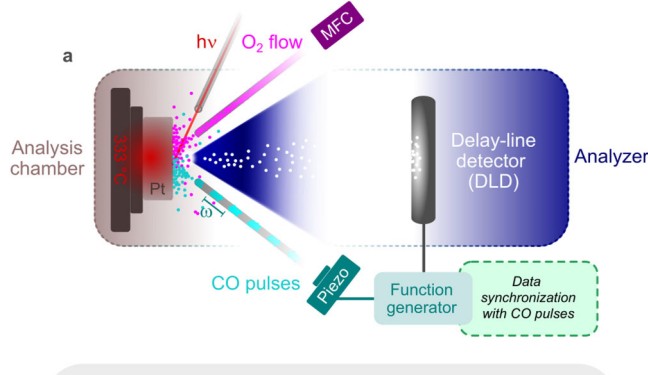

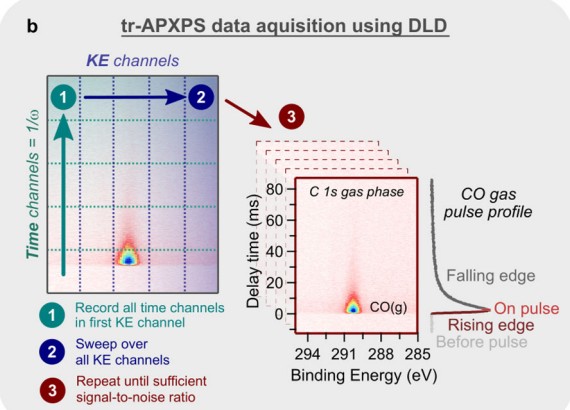

**Fig. 1 | tr-APXPS experimental set-up with data acquisition flow using a 2D delay-line detector. a** $O_2$ and CO gases are separately introduced facing a Pt(111) crystal in the analysis chamber. $O_2$ flow is set by a mass flow controller (MFC) and CO is pulsed at a specific frequency $\omega$, using a piezo valve synchronized to a delay-line detector (DLD) via a function generator. Using this set-up, event-averaged 2D plots of XPS data with a time axis defined by the pulsing frequency (1/$\omega$) are obtained. **b** Simplified visualization of tr-APXPS data acquisition where each time channel is measured at one kinetic energy (KE) channel before sweeping over another kinetic energy channel and measuring all time points. This procedure repeats until all kinetic energy channels are measured. Data collection continues as described until a sufficient signal-to-noise ratio is reached as defined by the user. An example 2D plot of the C 1s gas phase is presented with a side view of the CO pulse profile detailing regions of interest.

probe only gas phase molecules. Surface transformations induced by the CO pulse are evident in these 2D plots (Fig. 2a–e) and lead to increases in $CO_2$ production. For instance, Fig. 2c shows O adsorbates quickly replaced by CO adsorbates (CO$_{ads}$) after the CO pulse (~0 ms) and a gradual return to an O-covered surface when CO is pumped away (30–40 ms).

To assess surface changes linked to enhanced $CO_2$ production, tr-APXP line spectra were extracted, averaged accordingly to achieve a reasonable signal-to-noise ratio within each region, and fitted (Fig. 2f–j; see Methods). The fitted binding energies match well with the literature values (Supplementary Table 2). Figure 3 compares fitted areas at incremental time steps across the CO pulse to provide a comprehensive view of the reaction mechanism and enable the determination of active species driving $CO_2$ production by directly correlating gas phase product formation with surface changes (Fig. 3).

The O $1s$ GP signals in Fig. 3a illustrate the response of both reactants (CO, $O_2$) and products ($CO_2$); these are found to be in line with the C $1s$ GP signals (Supplementary Fig. 2). $CO_2$ forms immediately after CO is introduced, persists for 480 µs, then abruptly stops. As CO levels diminish on the falling edge, $CO_2$ production resumes after 2 ms and peaks after another 26 ms (IV), before it gradually declines over the falling edge. To understand these changes, we analyze the

response of the surface species−carbon and oxygen adsorbates−and the Pt catalyst.

In Fig. 3b, the C $1s$ areas show an absence of carbon adsorbates before the pulse, followed by the rapid growth of CO adsorbates (CO$_{ads}$) in two different configurations: CO bonded to one (CO$_{top}$) and two (CO$_{bridge}$) Pt atoms on the rising edge. $CO_2$ formation closely correlates with an increase in CO$_{ads}$ intensity on the rising edge of the pulse until saturation is reached after 400 µs. During the rest of the rising edge and the on-pulse region, CO$_{top}$ and CO$_{bridge}$ species remain relatively stable, with CO$_{top}$ at 61% of the total CO intensity (III). On the falling edge, CO desorption creates free sites for $O_2$ dissociation and subsequent reaction, triggering $CO_2$ production that peaks at IV, where the CO$_{ads}$ concentration is minimal and primarily composed of CO$_{top}$ (76%). The remainder of the falling edge continues to exhibit $CO_2$ production, but without CO$_{ads}$ signal being visible due to the low residence time of CO on the surface (likely less than thermal CO desorption at a few hundred µs[57]): surface-bonded CO reacts instantaneously with O adsorbates to form $CO_2$.

In Fig. 3c, O $1s$ areas trend shows both CO$_{ads}$ and O adsorbates signal with matching CO$_{ads}$ trends from the C $1s$ spectra (Fig. 3b). Prior to the CO pulse, the surface is covered by chemisorbed oxygen (O$_{chem}$) and Pt surface oxide (O$_{oxide}$). When CO is introduced on the rising edge of the pulse, O$_{chem}$ and O$_{oxide}$ are rapidly, within ~400 µs, replaced by CO$_{ads}$, creating a CO-covered surface. During this initial oxygen-rich phase, $CO_2$ production is high, indicating a highly catalytically active mixed O$_{chem}$−O$_{oxide}$ phase. Looking more closely at the time evolution of the two oxygen species shows that O$_{oxide}$ consumption is slightly delayed (120 µs) with respect to that of O$_{chem}$. As CO$_{ads}$ accumulate, the surface becomes poisoned, which hinders oxygen dissociation. This effect is reversed on the falling edge as CO gas is pumped away, reducing the CO$_{ads}$ equilibrium coverage and allowing oxygen to dissociate, forming O$_{chem}$ and O$_{oxide}$, which correlates with increased $CO_2$ production. Notably, the O$_{oxide}$ intensity increases significantly during the initial 34 ms of CO desorption/32 ms of $CO_2$ production, while the small amount of O$_{chem}$ intensity only markedly increases after $CO_2$ production peaks and O$_{oxide}$ stabilizes (red dashed box). The delayed increase of O$_{chem}$ intensity compared to that of O$_{oxide}$ during the falling edge mirrors the trends seen during the rising edge, suggesting a similar mechanism. This key observation implies that O$_{chem}$ plays an active role in CO oxidation, a point further explored in the "Discussion" section.

In Fig. 3d, the Pt $4f_{7/2}$ intensity trends demonstrate the response of the Pt catalyst. These reflect trends similar to those seen for the CO$_{ads}$, O$_{chem}$, and O$_{oxide}$ components in the O $1s$ spectra. On the rising edge, O$_{chem}$ reacts at a higher rate than O$_{oxide}$, a pattern mirrored on the falling edge, consistent with the above O $1s$ observations. Further analysis of the Pt $4f_{7/2}$ intensities shows an unexpected increase in coordinately unsaturated Pt atoms (Pt$_{surf}$) as O adsorbates are removed, implying exposed areas of bare metallic Pt. The Pt$_{surf}$ intensity peaks when O$_{chem}$ is fully consumed after 280 µs. At this stage, the remaining O$_{oxide}$ is consumed, and CO$_{ads}$ build on the surface with the depletion of Pt$_{surf}$. At CO$_{ads}$ saturation, Pt$_{surf}$ signal declines exponentially, potentially a result of slow Pt atom diffusion originating from Pt atoms/clusters that are left behind when CO$_{ads}$ react with oxygen atoms in O$_{oxide}$. Moreover, CO$_{bridge}$ signal increases as Pt$_{surf}$ signal decreases from 0.4–0.8 ms, suggesting that CO molecules are preferentially adsorbing in a bridged configuration as Pt sites become available. Pt$_{surf}$ signal continues to decline until ~1.2 ms but can be considered within the noise of the fit from 0.8–1.2 ms. This behavior repeats on the falling edge, with Pt$_{surf}$ decreasing as O$_{chem}$ builds up (after IV). Overall, tr-APXPS provides a comprehensive mechanistic explanation of gas phase and surface species dynamics throughout the CO oxidation reaction, allowing us to pinpoint periods of highest $CO_2$ production and understand the corresponding surface responses critical for its formation.

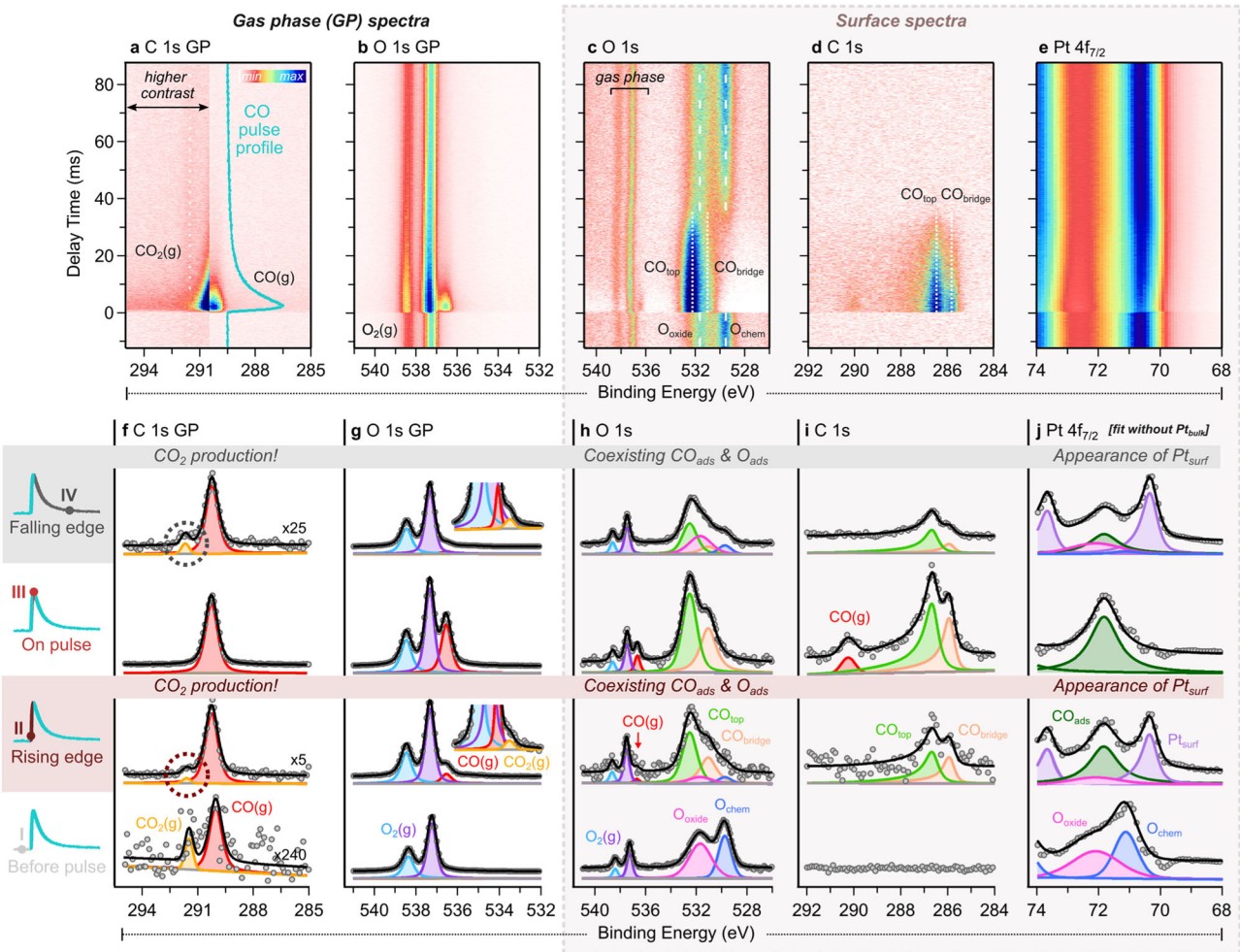

**Fig. 2 | tr-APXPS data of CO oxidation over Pt(111) with 40 µs time resolution when pulsing CO gas at 10 Hz into a constant flow of O₂ gas while monitoring gas phase and surface core levels.** 2D plots (top panel) and representative curve fit (bottom panel) at specific instances in time across the CO pulse as indicated by points I, II, III, and IV. Shown are the C 1s gas phase (GP) signal, at high binding energy displayed using a higher contrast region to amplify the faint $CO_2(g)$ feature (**a, f**), O 1s GP signal with insets magnifying CO(g) and $CO_2(g)$ when appropriate (**b, g**), O 1s (**c, h**), C 1s (**d, i**), and Pt $4f_{7/2}$ core levels (**e, j**). To accurately assess the Pt $4f_{7/2}$ fits (**j**), the $Pt_{bulk}$ component was removed (see Supplementary Fig. 1 for the original fit with $Pt_{bulk}$). Each row of fitted spectra (I–IV) equates to a slice of time on the 2D plots and represents significant points along the CO pulse within each pulsing region, also denoted by dashed lines in Fig. 3.

## Tuning catalytic activity with CO pulsing parameters

To further explore the CO oxidation mechanism, we focused on regions of high catalytic activity and inactivity by adjusting our pulsing scheme to mimic each independently. To target the peak reactivity region (IV in Fig. 3a), we tuned the pulsing parameters to aim at continuous $CO_2$ production resulting in pulsing CO at 50 Hz (Fig. 4). Increasing the pulsing frequency from 10 Hz to 50 Hz reduced the time window from 100 ms to 20 ms, leading to more gas overlap due to the limited pumping speed, resulting in the constant presence of CO. As shown in Fig. 4a, the C 1s GP spectra display continuous CO and $CO_2$ signals. The O 1s line in Fig. 4b mirrors the surface transformations seen at 10 Hz (Fig. 2c,h), with $CO_{ads}$ and O adsorbates alternating in response to CO pulsing, driving $CO_2$ formation. Despite a lower signal-to-noise ratio, it is still possible to fit the O 1s and C 1s data in coarse time steps (1 ms/50 spectra) to track surface composition changes. Given the intensity of the $CO_{ads}$ signal in the C 1s spectra (Fig. 4c), the C 1s GP and O 1s intensity trends in Fig. 4d highlight a preference for consumption of $O_{chem}$ over that of $O_{oxide}$ upon CO pulsing. Additionally, we observe a switch between $CO_{top}$ and $O_{chem}$: initially, at the start of the pulse, intensities are low for $CO_{top}$ and high for $O_{chem}$, but they become high for $CO_{top}$ and low for $O_{chem}$ after the pulse (Fig. 4d). This trend, reversed at 10 Hz on the falling edge in Fig. 3c, albeit with

different coverages, shows how 50 Hz targets the peak $CO_2$ production conditions seen at 10 Hz. This demonstrates the precision in controlling reactivity using CO pulsing parameters based on the structure-activity relationship.

Alternatively, we can adjust our pulsing parameters to deactivate the catalyst by extending the opening time of the CO pulsing valve to increase CO pulse pressure. $CO_2$ was not detected, and previous surface transformations in O 1s are no longer observed, replaced instead by a constant CO-rich environment (Supplementary Fig. 3). Comparison of traditional steady-state conditions to a modulating gas atmosphere also shows a level of reactivity control due to changes in surface interactions (Supplementary Note 1 and Supplementary Fig. 4).

## Discussion

The results indicate that favorable conditions for $CO_2$ conversion occur within narrow periods during CO pulsing, as the system transitions through low-activity CO- and O-poisoned phases. These active periods of $CO_2$ production occur on the rising and falling edges (and during the 50 Hz experiment), where surface transformations expose bare $Pt_{surf}$ areas, fostering reactive $O_{chem}$ and $CO_{ads}$ along with the formation of less reactive $O_{oxide}$.

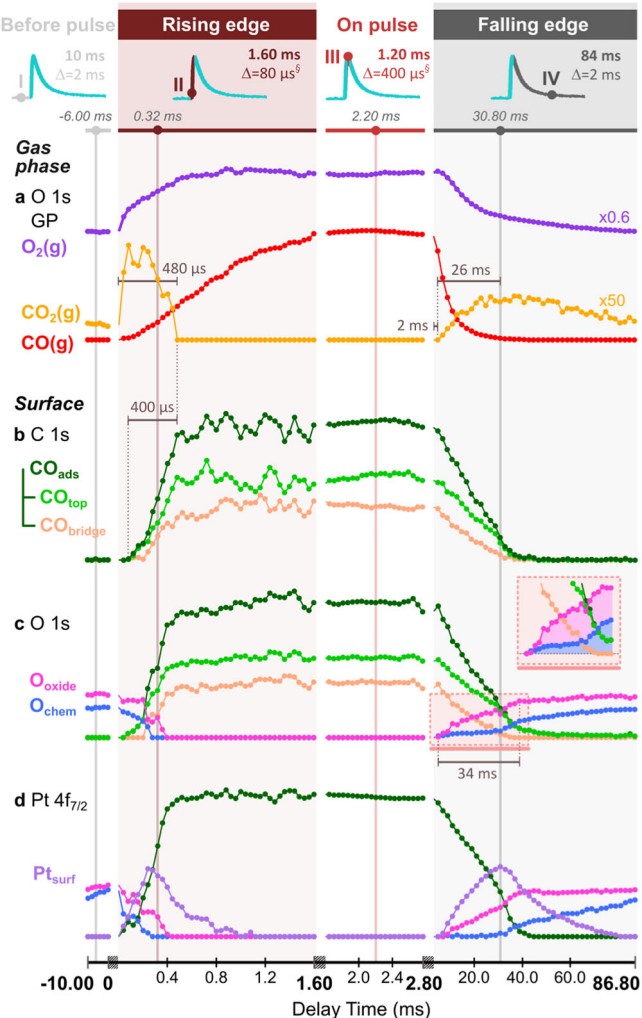

**Fig. 3 | Areas trend data comparing gas phase and surface species in relevant core levels and across delay time in the CO oxidation of Pt(111) when pulsing CO at 10 Hz into a stream of $O_2$ gas.** The analysis includes gas phase species determined from (**a**) O 1s core level, surface species as determined from (**b**) C 1s, (**c**) O 1s, and (**d**) Pt $4f_{7/2}$ spectra. **c** O 1s contains a red dashed box on the falling edge detailing a significant finding in the difference between chemisorbed oxygen (blue) and Pt surface oxide (pink) contributions. To effectively observe the evolution of all species throughout the reaction, different averaging of the spectra was required in each pulsing region as denoted at the top with or without a running average (§). Consequently, the noise level changes across pulsing regions. Physical breaks between pulsing regions are employed to help distinguish different regions. Particularly meaningful slices of time in each pulsing region are conveyed by guide lines with associated roman numerals (I–IV) whose fits can be found in the bottom panel of Fig. 2.

The most plausible explanation for the observed trends is that $O_{chem}$ is the primary active species. The lack of $O_{chem}$ signal alongside increasing $O_{oxide}$ and $CO_2$ production on the falling edge is explained precisely by the high activity of $O_{chem}$, which is rapidly consumed to form $CO_2$ (Fig. 5). $O_{oxide}$ begins to form when the *local* $O_{chem}$ concentration exceeds 0.25–0.50 monolayers[58–62], appearing in patches of bare Pt where $O_{chem}$ and $CO_{ads}$ react. This less reactive $O_{oxide}$ accumulates on the surface and can react with CO, as observed on the rising edge of the CO pulse, albeit at a lower rate[13,35,63,64]. When $CO_2$ production lags, the $Pt_{surf}$ concentration decreases as $O_{chem}$ increasingly occupies the surface, as observed in its enhanced signal. The system then moves toward a thermodynamically favored state with the stabilization of $O_{oxide}$ shortly thereafter and little reactivity towards $CO_2$

conversion[58]. In agreement with these observations, the increased activity of $O_{chem}$ causes larger oscillations compared to $O_{oxide}$ during the 50 Hz experiment. Such a classical LH mechanism is supported by many experimental studies demonstrating the lack of $O_{oxide}$ or its lower activity in the reaction[8–14]. This reaction was shown to predominantly occur on the terraces, with the step edges playing a minor role at the temperature used in the study[65–67]. While $O_{oxide}$ is detected in large amounts on the active surface, $O_{chem}$ drives the reaction, making it a minor yet crucial species due to its transient nature[68].

Nonetheless, the less reactive $O_{oxide}$ likely contributes slightly to $CO_2$ conversion, as observed on the rising edge where $O_{oxide}$ is consumed at a slower rate than $O_{chem}$. We propose a secondary CO oxidation mechanism involving $O_{oxide}$, consistent with presented experimental data and previous findings, such as those that have predicted higher activity *at the boundary* between the metallic (CO-covered) and oxide phases[34,69–75]. This mechanism requires the formation of individual $PtO_4$ clusters prior to the start of the catalytic reaction when the *local* oxygen coverage reaches 0.25–0.50 monolayers within bare Pt patches (as described earlier). Cluster formation has an estimated activation barrier of 0.44-0.82 eV from a pre-existing chain of $O_{chem}$[60] or three individual $O_{chem}$[50], respectively, comparable to $O_{chem}$ diffusion (0.63 eV)[50] or phase-dependent $CO_{ads}$ oxidation (0.49-1.01 eV)[13,64,76–79]. Further cluster growth into chains is hindered by a large activation barrier (1.13 eV)[50], allowing $O_{chem}$ and $O_{oxide}$ to coexist in close proximity until they react with $CO_{ads}$. Under the present experimental conditions of gas pressure, temperature, and exposure time, we can infer that we likely have a 4O-type[14] surface oxide ($O_{oxide}$) that coexists with $O_{chem}$ on metallic Pt, based upon the expansive literature surrounding oxidation of Pt and identification via O 1s and Pt 4f binding energies[80–84]. Bulk or surface α-$PtO_2$ trilayer oxide appear at a much higher binding energy of 73.5–74.1 eV and requires higher $O_2$ pressure and temperature[60,64,85,86]. In equilibrium, this oxide forms stripes with platinum atoms raised by 1.7 Å, each surrounded by four oxygen atoms, but likely beginning with smaller cluster-type structures similar to the $PtO_4$ clusters needed for the proposed reaction mechanism[58]. $Pt_{surf}$ binding energy trends suggest the Pt surface transitions from a (111)-type surface to a more step-like structure (Supplementary Fig. 5). This supports the proposed dual mechanism: $CO_{ads}$ initially reacts with $O_{chem}$, exposing bare Pt, but as $O_{oxide}$ continues to grow, it dominates the reaction, leading to a stepped surface appearance. Furthermore, the diminished $Pt_{surf}$ intensity with the stabilization of $O_{oxide}$ may contribute to the lack of reactivity based upon the observed inactivity on a similar oxygen-terminated surface oxide on Rh(111) due to missing coordinately unsaturated sites[87] as compared to highly reactive PdO(101)[88] and $RuO_2$(110)[89] with available coordinately unsaturated sites.

The catalytic reaction starts when $CO_{ads}$ abstracts a single oxygen atom from an individual oxide cluster to produce $CO_2$, a less activated step (0.1–0.3 eV)[64,66] than the reaction with $O_{chem}$. If the remaining partially reduced cluster is less reactive than *both* the initial cluster *and* $O_{chem}$, it remains stable until a $O_{chem}$ atom attaches to the cluster, closing the catalytic cycle. This attachment is facile with a low activation energy (0.08–0.26 eV)[50], lower than the barrier for $O_{chem}$ diffusion or reaction with $CO_{ads}$. For this reaction step, the availability of $O_{chem}$ atoms in the immediate vicinity of the oxide is critical. The high reactivity of oxide clusters ensures bare Pt sites for $O_2$ dissociative adsorption, supplying $O_{chem}$ directly next to the clusters. Thus, the most catalytically active surface for oxide clusters is composed of a largely CO-free Pt surface with numerous bare metallic Pt atoms readily available for $O_2$ adsorption and dissociation[90]. In CO-rich environments, the perimeter of partially reduced clusters remains CO-covered, preventing re-oxidation to a highly active state. Whereas O-rich environments promote the formation of thermodynamically stable but less active oxides, e.g. trilayer $PtO_2$. We also do not observe carbonate intermediates, theoretically predicted for direct oxidation of CO on trilayer

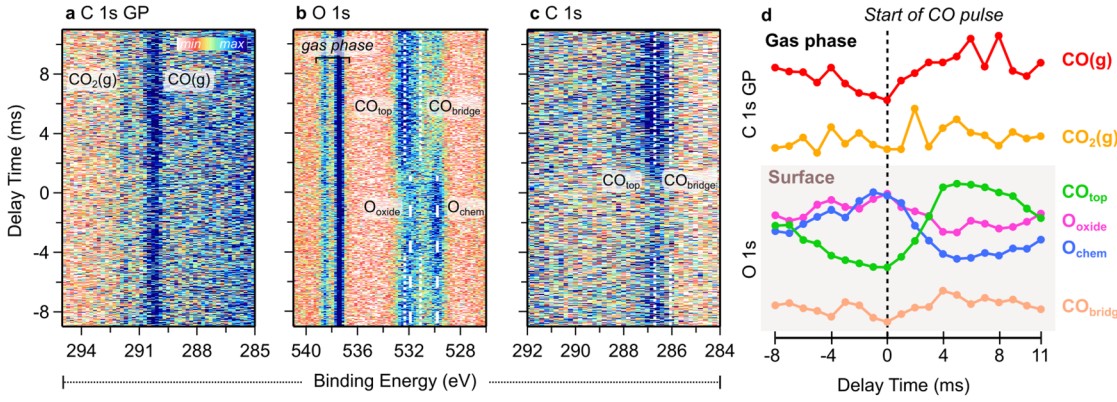

**Fig. 4 | tr-APXPS of CO oxidation of Pt(111) when pulsing CO at 50 Hz into a stream of $O_2$ where $CO_2$ production can be controlled by tuning the pulsing parameters thus changing the surface-mediated response.** Constant $CO_2$ production was achieved as shown in 2D plots of the **a** C *1s* GP, **b** O *1s*, and **c** C *1s* spectra, along with an analysis of **d** O *1s* intensity trends.

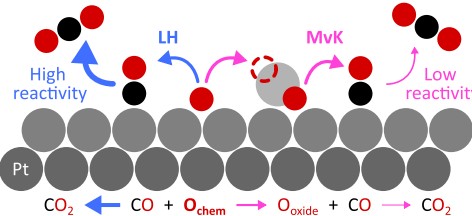

**Fig. 5 | Proposed reactivity scheme of CO oxidation of Pt(111).** High $CO_2$ production is emphasized upon reaction of $CO_{ads}$ with $O_{chem}$ as opposed to surface oxide $O_{oxide}$. $O_{chem}$ primarily reacts with $CO_{ads}$ to produce $CO_2$ (LH mechanism) but also forms $O_{oxide}$. In turn, $O_{oxide}$ occasionally reacts with $CO_{ads}$ to form $CO_2$ but is much less reactive than $O_{chem}$ (MvK mechanism).

$PtO_2$ indicating that the surface oxide is not active[91–93]. The proposed difference in activity between a complete and partially reduced oxide cluster explains the discrepancy between the calculated activity of the 4O phase ($O_{oxide}$) and experimental findings, such as those by Miller et al.[14], which showed lower reactivity than $O_{chem}$. If only one of the two oxygen atoms in the 4 O phase is catalytically active, the overall reactivity of the phase will be limited by the abstraction of the second oxygen atom with potentially higher activation energy.

Moreover, the results reveal the chemical transformations behind the $CO_2$ spikes commonly detected at the start of oxidation/reduction pulses in a modulated gas feed, as used in industry[26,42,44,46]. Periodic modulations of the gas feed force the surface to oscillate between CO-poisoned and O-poisoned states, intermittently passing through the catalytically active phase described above. This also clarifies discrepancies in previous conflicting reports, where observations under steady-state conditions could favor different surface chemistry that does not capture transient species.

Altogether the optimized reaction conditions in this experiment concerning temperature, pressure, and $CO/O_2$ ratio provided the means to target a highly reactive state and bridge the pressure gap, as confirmed by the observation of a surface oxide. Altering these parameters would likely change the response of the catalyst by targeting different regions of the phase diagram with temperature and pressure changes. For example, a lower reaction temperature could lead to CO poisoning throughout the experiment, inhibiting oxygen adsorption, and prohibiting $CO_2$ production. Higher pressure could promote bulk oxide formation, likely accompanied by a different reaction mechanism than the current CO oxidation study involving chemisorbed oxygen and surface oxide. The initial $CO/O_2$ ratio has less of an effect on the reaction due in large part to our pulsing scheme where the $CO/O_2$ ratio is inherently changing throughout the CO pulse, constantly

ramping over inactive and reactive states. We observe this effect in Supplementary Fig. 4 where the $CO_2(g)$ signal remains stable when changing the initial $CO/O_2$ ratio from 7.5/1 to 30/1.

In conclusion, the present tr-APXPS results suggest a primary LH mechanism involving $O_{chem}$ as the active species with a secondary MvK mechanism concerning less reactive, largely spectator $O_{oxide}$ clusters. The reaction takes place at specific instances during the CO pulse instigating open Pt areas for $O_2$ dissociation into reactive $O_{chem}$ immediately consumed by a) $CO_{ads}$ to produce $CO_2$ and b) formation of $O_{oxide}$. These observations offer new insights into the highly contested CO oxidation mechanism. The incorporation of both operando conditions and microsecond time resolution in tr-APXPS enabled the discovery of these highly active, transient species produced during surface transformations, facilitating the concurrent detection of reaction products, surface intermediates, and catalyst response. Even better time resolution could be possible in the future with a faster piezo valve to match the nanosecond time resolution of the DLD. We demonstrate how tuning pulsing parameters in periodic operation can optimize $CO_2$ formation, offering a valuable tool for improving catalytic technology designs.

The presented findings provide a strong motivation for revisiting the proposed mechanisms of some catalytic reactions; especially in cases when several distinct species can be attributed to the active component. Comparing the pathways of CO oxidation on hexagonal surfaces of other Pt-group metals is a natural next step. On the other hand, the time resolution is an additional benefit in chemical analysis of surfaces of novel catalytic materials.

## Methods

### Sample preparation and acquisition of time-resolved data

All experiments were performed at the HIPPIE beamline of the MAX IV Laboratory in Lund, Sweden[94]. Sample preparation took place on the solid-gas endstation (SGE). A Pt(111) single crystal mounted on a flag-style plate with a K-type thermocouple spot-welded to the sides of the crystal underwent two sputter (20 min at 1 kV and 10 mA emission current in $1 \times 10^{-5}$ mbar Ar) and anneal (10 min at 600 °C) cycles. Pt was then transferred via air from SGE to the solid-liquid endstation of HIPPIE for tr-APXPS measurements where the backfilled chamber set-up has a routine base pressure of -5 × 10⁻⁶ mbar when pumped with a 300 L turbomolecular pump. The endstation uses a SPECS PHOIBOS-NAP spectrometer equipped with a Surface Concepts 2D delay-line detector (DLD), which captures the arrival time of photoelectrons hitting its delay lines.

During the experiment, Pt(111) was kept at a reaction temperature of 333 ± 2 °C (as monitored by the thermocouple). The initial $O_2$ pressure of -3.0 × 10⁻³ mbar burned away any residual carbon as verified by XPS. $O_2$ and CO gases had separate gas lines whose valves were

installed on the outside of the chamber with internal piping attached to the cone directed at the face of the sample (a few mm away). $O_2$ gas (N5.5 purity) was supplied via a mass flow controller set to 30 sccm throughout the experiment, resulting in a ~ $3.0 \times 10^{-3}$ mbar pressure reading. CO gas (N3.7) was pulsed into the chamber via a Attotech GR020 piezo valve providing mbar internal pressure[56]. Note that the close proximity of the gas pipes at the front of the sample resulted in $O_2$ gas entering the CO pipe between CO gas pulses and was subsequently pushed out when pulsing as observed in an increase in $O_2$ gas phase signal (Figs. 2b and 3a). Gas attenuation may also play a role as photoelectrons will scatter less with CO than $O_2$ leading to an increase in $O_2$ signal with the introduction of CO. The operations of the piezo valve and the DLD were synchronized using a BNC 565 function generator (Fig. 1). The CO gas line contained a filter (PAL) to purify the gas and was constructed mainly from Cu pipe to minimize the well-known Ni carbonyl contaminates that form under elevated CO pressure upon reaction with stainless steel. The Ni 2p line was monitored continually throughout all experiments. The piezo valve required a constant back pressure of ~3 bar to ensure stable pulsing conditions. CO pulsing parameters varied depending on the experiment. Figures 2 and 3 and Supplementary Figs. 1, 2, and 5 represent datasets where the piezo valve was pulsed at 10 Hz under 240 V and 120 μs opening time. The pulsed data in Supplementary Fig. 4 were also recorded at 10 Hz, while varying the voltage (170–240 V) to obtain different $O_2$:CO ratios. Figure 4 and Supplementary Fig. 3 show datasets where the piezo valve was pulsed at 50 Hz, 160 V, and 53 μs vs. 170 V and 250 μs, respectively.

Using a special measurement mode in the SPECS Prodigy software connected to the timing hardware in the analyzer and external synchronization of CO gas pulses with the DLD provided the means to acquire time-resolved data while sweeping over kinetic energy channels to acquire 2D plots of binding energy vs. time. The time axis was derived from the CO pulsing frequency and number of time channels: CO pulsing at 10 Hz and 2500 channels provided a 100 ms time window with 40 μs time resolution, whereas CO pulsing at 50 Hz and 500 channels provided a 20 ms time window with 40 μs time resolution. While our previous studies[56] used a simple pump-probe measurement scheme measuring one delay at a time, we measured all delays simultaneously with the upgraded detection scheme from camera to DLD (Fig. 1b).

Core levels of interest included the O $1s$ (hv = 750 eV), C $1s$ (750 eV), and Pt 4f (260 eV) lines, recorded at a pass energy of 100 eV with a 0.8 mm analyzer slit. O $1s$ and C $1s$ spectra were acquired at both gas phase and surface positions, whereas the Pt 4f line was measured only in the surface position. As adsorbed CO is always located on the surface, we did not tune the photon energy to be surface-sensitive for acquiring C $1s$ spectra. The surface position describes the typical XPS sample position 300 μm away from the analyzer cone (cone diameter 300 μm). The gas phase position is 450 μm from the standard XPS position; here, only gas phase species are detected. Note that the APXPS detection of gas phase species is better than conventional mass spectrometry (MS): First of all, it benefits from the synchronization of gas injections and high time resolution of the DLD. Secondly, it probes the local gas composition in the near vicinity of the sample surface (as detailed in Supplementary Fig. 6 of Küst et al.[95]). Therefore, MS data was not recorded in the present study. The beamline exit slit was open to 50 μm at the surface position and, 100 μm and 200 μm for O $1s$ and C $1s$ gas phase positions, respectively.

### Data analysis

tr-APXPS data was analyzed in IgorPro (WaveMetrics) and fitted using the EccentricXPS procedure. Each 2D plot was energy-calibrated against the Fermi level. Linear, Shirley, and polynomial background subtractions were administered depending on the core level and sample location. All gas phase spectra (O $1s$, C $1s$) required a linear background subtraction. Pt 4f and C $1s$ spectra required Shirley

background subtraction. Deviating from that general procedure, a linear background was subtracted from the C $1s$ spectra when the CO adsorbate signal was not detected (i.e., before and after the CO pulse). The O $1s$ line required additional care due to its close proximity to the Pt $4p_{3/2}$ level, necessitating a fourth-order polynomial fit to accurately approximate the tail of the Pt feature. In the presence of high-pressure CO from the CO pulse, a Shirley background subtraction was necessary before the polynomial background subtraction. Post background subtraction, all spectra were fit with Voigt peak shapes compounded with an asymmetric factor when needed, e.g., Pt 4f bulk and surface features. All Pt 4f spectra were intensity normalized by the total Pt peak area.

Further processing of tr-APXP spectra included averaging spectra differently in the four pulsing regions to achieve a reasonable signal-to-noise ratio while preserving relevant time information. For instance, the rising edge contains essential information about the surface response when the CO pulse is first introduced, requiring minimal averaging to capture compositional changes within this short timeframe, i.e., 80 μs averaging over 2 spectra. Conversely, the falling edge has a slower response time due to the dependence of CO removal on pumping speed, leading to 2 ms averaging over 50 spectra for the 10 Hz dataset. The before-pulse region followed the same treatment as the falling edge since it is the tail of the falling edge, whereas the on-pulse region required averaging over 400 μs/10 spectra for the 10 Hz dataset. A running average with a step of 1 was also performed on the rising edge and on-pulse spectra to ensure all reaction intermediates were captured with sufficient time resolution. Note that the 50 Hz dataset required averaging over 1 ms/50 spectra to achieve a reasonable signal-to-noise ratio for fitting. All fitting parameters can be found in Supplementary Table 3.

### Comparison between pulsing and flow experiments

CO pulsing vs. flow experiments required adjusting either the CO mass flow controller (4 sccm) or CO pulsing voltage (170–240 V), respectively, to achieve the targeted CO:$O_2$ ratios, while $O_2$ remained at 30 sccm output throughout the experiments and the system heated to 333 °C. While monitoring the response of CO(g) in C $1s$, the CO pulsing voltage was adjusted to match the intensity of the CO(g) peak measured in the flow experiment, thus ensuring the same CO:$O_2$ ratio when pulsing. All spectra were collected in time-resolved mode and integrated to produce the line spectra presented in Supplementary Fig. 4.

## Data availability

The XPS data generated in this study have been deposited in the Figshare database [https://doi.org/10.6084/m9.figshare.28067600].

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

## Acknowledgements
C.N.E., W.W., J.P., R.H.T., M.S., J.S., J.K. and A.S. acknowledge MAX IV Laboratory for time on HIPPIE under Proposals 20230723 and 20230835. Research conducted at MAX IV, a Swedish national user facility, is supported by the Swedish Research Council (VR) under contract 2018-07152, the Swedish Governmental Agency for Innovation Systems VINNOVA under contract 2018-04969, and Formas under contract 2019-02495. UK and JK acknowledge financial support from the Swedish Research Council (2022-04363) and the Crafoord Foundation. We acknowledge Edvin Lundgren for discussing the results and proposed reaction mechanisms.

## Author contributions
C.N.E. acquired the data, analyzed and interpreted the results, and drafted and revised the manuscript. W.W. designed and built the experimental hardware and acquired the data. U.K. and J.P. acquired the data and drafted the manuscript. R.H.T. and M.S. supported the experiments and drafted the manuscript. J.S. drafted the manuscript. J.K. designed the experiment, acquired the data, and drafted and revised the manuscript. A.S. designed the experiment, acquired the data, interpreted the results, and drafted and revised the manuscript.

## Funding

## Competing interests
The authors declare no competing interests.
