## [Peer Review file · Nature Communications]

Resolving Active Species during the Carbon Monoxide Oxidation over Pt(111) on the Microsecond Timescale

Corresponding Author: Dr Andrey Shavorskiy

Version 0:

Reviewer comments:

Reviewer #1

(Remarks to the Author)

The study by Eads et al. employs a time-resolved APXPS method to address debates on reaction mechanism of CO oxidation over Pt(111). This innovative design aids to differentiate between active and spectator species under reaction conditions, which is creative. However, more robust data should be provided to support the conclusion, and certain areas require further explanation and refinement:

- (1) The authors may introduce more about the setup since this is the main difference from other works. Furthermore, the presented data in the paper are not very rich. The authors should present more results for example data on well-known CO oxidation on Ru(0001) surface for comparison.
- (2) I disagree with the statement "tuning pulsing parameters in periodic operation can optimize CO₂ formation, offering a valuable tool for improving catalytic technology designs". While the gas pulse method is effective for investigating reaction mechanisms, particularly the interactions between reactants with different adsorbates, it may not be a practical method for enhancing catalytic activity in real reaction conditions. Thus, the claim "offering a valuable tool for improving catalytic technology designs" is not proper.
- (3) The conclusion in this work is insufficiently rigorous and should specify the boundary conditions. The reaction mechanism depends on temperature, pressure and CO/O₂ ratio. Would it be accurate to state that varying the temperature (both lower and higher) and pressure (noting that the pressure used here is significantly low) affects the results
- (4) Do the authors acquire MS spectra simultaneously with the XPS spectra? MS results would provide additional information about the reactants and products.
- (5) The kinetic energy of emitted electron from O 1s and Pt 4f is set as ~ 200 eV. Why ~500 eV for C 1s?
- (6) It is interesting to observe that during the falling edge, Pt surface oxide is formed while chemisorbed oxygen species are nearly absent. Does this imply that Pt reacts directly with O₂ to form surface oxide rather than through chemisorption followed by reaction?
- (7) The authors are encouraged to make the 2D plots and spectra more succinct and clearer to help readers understand the changes in different species.

Reviewer #2

(Remarks to the Author)

In this study, Eads et al. investigated CO oxidation on Pt(111) using tr-APXPS to distinguish between active catalytic species and spectator species under millisecond time scales. By modulating CO pulses in an O₂ stream, they identified chemisorbed oxygen as the primary reactive species responsible for CO₂ formation, while the surface oxide primarily acted as a spectator. Although this study confirms the well-established Langmuir-Hinshelwood (LH) mechanism, it significantly enhances our understanding of the structure-activity relationship and enables dynamic control of CO₂ production through CO pulsing. Therefore, while this study is suitable for publication, the following points must be addressed first.

1. These measurements were conducted at 333°C, which is well above the typical temperature range for CO oxidation reactions on Pt surfaces. Are these findings relevant to the species present on the Pt surface—such as the oxide phase, adsorption sites, and the coverage of chemisorbed oxygen and CO—under more realistic reaction temperature conditions?
2. In Figure 2a, O₂ GP peak seems to be increasing during rising edge and decreasing during falling edge. The reason of this behavior needs clarification.
3. Oxide terms and their surface coverage that is used in this study is not well-defined. Reactivity difference with respect to

chemisorbed O is clear. In discussion section the authors hypothesize PtO₂ and Pt-O cluster formation, however the type of oxide before the pulse, rising edge and falling edges need better description.

4. Is the type of the oxide forming during falling edge the same as the as initial one? It is interesting that oxide appears before the chemisorbed oxygen in falling edge. What would be the mechanism of the formation of this oxide phase in the presence of adsorbed CO and free Pt?

5. MvK mechanism involves diffusion of oxide ions from the bulk, perhaps diffusion on the surface could be possible. What is the evidence that removal of the oxide follows MvK mechanism? One can also argue that adsorption rate of CO and its sticking coefficient is different on or near oxide phase. Moreover, the adsorption site of CO in MvK mechanism of the oxide phase is not clear.

6. Ptsurf remaining until 1.2 ms (Figure 2d) despite surface already dominated by CO is not realistic. Could this be a fitting error? The authors must comment on this.

7. Minor comment: On page 3, the authors report that they monitor surface and gas phase evolution below 1 ns time resolution. I believe this is a typo. If this has something to do with DLD counting, I think it requires rephrasing since the time resolution of this work is not in ns range.

8. Minor comment: The color or position of CO_{top} and CO_{bridge} in Figure 1c should be adjusted to avoid confusion with chemisorbed O during the falling edge.

Reviewer #3

(Remarks to the Author)

This is an important work, on one hand reporting the implementation for the first time of efficient pump-and-probe Ambient Pressure X-Ray photoelectron Spectroscopy (APXPS) using periodical chemical perturbations -gas pulses- to achieve sub-millisecond time resolution, and on the other hand using this innovation to cast light on the detailed mechanisms of the thoroughly investigated CO oxidation reaction on Pt(111), with potential impact also on the optimization of dynamic catalyst operation.

Compared to a previous publication from the same group, ref. 56, corresponding to a preceding stage of their time resolved APXPS project, now the method achieves its full potential with the upgrading of the the camera to a DLD detector, which enables acquiring all the time delays for each kinetic energy in a single gas pulse, increasing the efficiency by one or two orders of magnitude.

The method enabled the detection of O_{chem} and O_{oxide} species, and their time correlation with high and intermediate CO oxidation activity, respectively, making it possible to deduce the coexistence of the two proposed reaction mechanisms, LH and MvK. This is relevant not only from a fundamental perspective, as it allows for nicely harmonizing the broad amount of information on this reaction, but it also has great significance in the understanding and optimization of industrial CO oxidation using pulsed gas.

In summary, this is an outstanding work, and I recommend its publication in Nature Communications after consideration of the following minor comments:

a) It would be useful to detail how many CO pulses and total acquisition time were needed for the data in Fig. 1.

b) At first sight, the non-linear time scale of Fig. 2, makes one -or me- believe that the surface was most of the time in CO-poisoned state. This additionally leads to a difficulty understanding why the CO pulsing frequency was in a later experiment increased to 50 Hz (if the surface was already poisoned at 10Hz, I thought). While it is true that in Fig. 1 it is clearly seen that it is not the case that at 10 Hz the surface is most of the time poisoned, it might still be useful to call attention in the text or in the caption to the non-linear time scale in Fig. 2.

c) In the pulse vs flow experiments reported in the Supplementary Note 1, was the temperature also T=333C?

d) In line 105: “.. we revolutionized”, I think that it should read: “we revolutionize”

e) In line 107: monitor the surface and gas phase evolution with high (below 1 ns)...”, it should be “... (below 1 ms)..”

f) In line 184: would it be possible to provide an estimate of the residence time of CO in the Pt(111) surface, rather than defining it as “extremely low” or “CO reacts instantaneously”? Is it of the order of ns?

g) In line 226 it is stated that the time step for fitting the O1s and C1s 50 Hz data was 1ms/50 spectra, however in the “Methods” section it is said that it was 280us /7 spectra.

Version 1:

Reviewer comments:

Reviewer #1

(Remarks to the Author)

I can recommend the publication in the current form.

Reviewer #2

(Remarks to the Author)

The authors have done a great job addressing my and other reviewers' comments. I believe this study is now suitable for publication in Nature Communications.

Reviewer #3

(Remarks to the Author)

The authors have satisfactorily addressed all my comments. I recommend the publication of the revised manuscript in Nature Communications.

The original reviewer's comment is displayed in black. Our response is highlighted in red. *The proposed modifications are highlighted in red italics incl indication in which part of the manuscript the changes are proposed. Finally, in the manuscript text, the proposed modifications are marked in red.*

NB! While revising the manuscript, we realized the exact formatting visible on the screen depends on the version of Word, specifically whether it is in Mac or Windows. In particular, the line numbers referenced below can be different depending on which platform the file is open. Unfortunately, we do not know the origin of the issue and could not fix it easily. However, the difference in line numbers is small (only 2-4 lines) and the new text is clearly highlighted in red in the manuscript. We hope, therefore, that this potential issue will not cause any challenges for the reviewers.

Reviewer #1 (Remarks to the Author):

The study by Eads et al. employs a time-resolved APXPS method to address debates on reaction mechanism of CO oxidation over Pt(111). This innovative design aids to differentiate between active and spectator species under reaction conditions, which is creative. However, more robust data should be provided to support the conclusion, and certain areas require further explanation and refinement:

(1) The authors may introduce more about the setup since this is the main difference from other works. Furthermore, the presented data in the paper are not very rich. The authors should present more results for example data on well-known CO oxidation on Ru(0001) surface for comparison.

We appreciate knowing that an account of the methodology is of interest to mention in the main text rather than fielded only in the Methods section and Supplementary Information. We reworked the beginning of the Results section to accommodate a detailed perspective of the methodology and presented an updated version of Figure 5 as the new Figure 1 to provide a visualization of the experimental set-up and how tr-APXPS data is obtained using a 2D delay-line detector.

"Time-resolved methodology. Contrary to a typical reactor scheme..." on p. 3 (l. 123-153).

Regarding the scientific case, the effectiveness of Pt in the CO oxidation reaction is unrivalled and countless studies have been performed using various methods to untangle why Pt is an ideal catalyst. This knowledge remains pertinent due to the desire to recreate a similar mechanism on more abundant catalyst materials in order to move beyond Pt. Our tr-APXPS approach provides an effective approach to do so by unveiling microsecond to millisecond electronic changes in surface intermediates that were previously ill-defined. Capturing this chemical information has allowed us to piece together two mechanisms in action with one (LH) more prominent than the other (MvK). This new information will hopefully settle the longstanding debate regarding the mechanism of CO oxidation over Pt and bring the field closer to supplementing Pt for other catalysts that can provide a similar mechanism to that of Pt. Thus, it is our belief that our study is quite self-contained and impactful focusing on Pt rather than diluting our message with further experiments on related catalyst surfaces, at least for this particular study. Our aspiration is for researchers to take hold of the tr-APXPS approach and explore more catalytic reactions using a variety of different catalysts.

We highlighted this belief in the last paragraph of the Discussion and Conclusions section on p. 10 (l. 386-390):

“The presented findings provide a strong motivation for revisiting the proposed mechanisms of some catalytic reactions; especially in cases when several distinct species can be attributed to the active component. Comparing the pathways of CO oxidation on hexagonal surfaces of other Pt-group metals is a natural next step. On the other hand, the time resolution is an additional benefit in chemical analysis of surfaces of novel catalytic materials”.

(2) I disagree with the statement “tuning pulsing parameters in periodic operation can optimize CO₂ formation, offering a valuable tool for improving catalytic technology designs”. While the gas pulse method is effective for investigating reaction mechanisms, particularly the interactions between reactants with different adsorbates, it may not be a practical method for enhancing catalytic activity in real reaction conditions. Thus, the claim “offering a valuable tool for improving catalytic technology designs” is not proper.

We agree that the described method would be impractical to directly apply to catalytic technology. In the form it was presented in the manuscript, it is, indeed, rather an excellent method for studying surface reactions. However, interestingly, some industries already implement a pulsing methodology in their design and perform research to optimize this scheme under normal operating conditions. For instance, CO oxidation taking place in automotive catalytic converters has an inherent modulation in gas atmosphere due to a feedback loop that sets an effective air-to-fuel ratio in the exhaust gas which constantly corrects fuel and air injections creating rapid transient changes in gas composition. Researchers at Toyota and General Motors performed numerous experiments testing modulation parameters on real catalysts under practical conditions and found differences in CO₂ product formation. Their studies showed an effect of catalyst response when changing modulation conditions, however, the reason remains unknown. Our experimental approach provides a window into the mechanistic aspects of the reaction, answering the question of *why* the catalyst performs as it does, which can lead into optimization efforts.

We included this information in the Introduction section on p. 2 (l. 84-89),

“For instance, automotive catalytic converters have an inherent modulation in gas atmosphere due to a feedback loop that sets an effective air-to-fuel ratio in the exhaust gas which constantly corrects fuel and air injections creating rapid transient changes in gas composition. Researchers at Toyota and General Motors performed numerous experiments testing these modulation parameters on real catalysts under practical conditions and found differences in product formation^{41,45}.”

(3) The conclusion in this work is insufficiently rigorous and should specify the boundary conditions. The reaction mechanism depends on temperature, pressure and CO/O₂ ratio. Would it be accurate to state that varying the temperature (both lower and higher) and pressure (noting that the pressure used here is significantly low) affects the results

Generally, in any catalytic reaction, temperature, pressure, and gas ratio will affect how the catalyst responds. Our approach still requires selecting conditions that work for CO conversion. For instance, using a low enough temperature would likely result in CO remaining on the surface preventing oxygen adsorption and dissociation thus blocking the pathway for CO₂ production. In our existing experimental set-up, increasing the gas pressure could lead to a CO poisoning effect due to the limited pumping capacity which would provide a higher CO background pressure. The results could be similar to the 50 Hz dataset where

CO pressure is sufficiently high at all times, thus propagating a CO poisoned surface. However, this issue could be solved with more efficient pumping and under these ideal conditions increasing gas pressure should not affect the results unless the surface itself is changing in a different way that is only triggered at a higher CO pressure on the catalyst. For instance, we are bridging the pressure gap with the observation of surface oxide under these conditions, however a bulk oxide could be reached at higher pressures. We show the impact of changing the CO/O₂ ratio in our pulsed vs. flow experiments presented in Supplementary Fig. 4 with an explanation in Supplementary Note 1. To summarize, the CO₂ area is fairly consistent throughout different CO/O₂ ratios (1/7.5 to 1/30) indicating a low sensitivity towards the initial CO/O₂ ratio compared to flow experiments. This is likely due to the constant change in CO/O₂ ratio throughout our pulsing experiment with the introduction of the pulse and when CO is being pumped away, sweeping through highly active states at certain CO/O₂ ratios on the rising and falling edges.

We summarized these points in the Discussion and Conclusions section on p.10 (l. 362-372),

“Altogether the optimized reaction conditions in this experiment concerning temperature, pressure, and CO/O₂ ratio provided the means to target a highly reactive state and bridge the pressure gap as confirmed by the observation of a surface oxide. Altering these parameters would likely change the response of the catalyst by targeting different regions of the phase diagram with temperature and pressure changes. For example, a lower reaction temperature could lead to CO poisoning throughout the experiment, inhibiting oxygen adsorption, and prohibiting CO₂ production. Whereas higher pressure could promote bulk oxide formation likely accompanied by a different reaction mechanism than the current CO oxidation study involving chemisorbed oxygen and surface oxide. The initial CO/O₂ ratio has less of an effect on the reaction due in large part to our pulsing scheme where the CO/O₂ ratio is inherently changing throughout the CO pulse, constantly ramping over inactive and reactive states. We observe this effect in Supplementary Fig. 4 where CO₂(g) signal remains stable when changing the initial CO/O₂ ratio from 7.5/1 to 30/1.”

We would like to stress that conducting additional experiments to investigate the effects of the Ru(0001) surface (R1, question 1) or varying pressure/temperature conditions would significantly delay the communication of our novel experimental capabilities to the scientific community. Additionally, incorporating extra sets of data into our analysis and discussion will dilute the impact of the current narrative and eliminate its clarity and brevity. In our view, it is more important to avoid unnecessary delays and to preserve the concise and impactful nature of our study. We strongly believe that the results of such investigations should be explored in future publications, either by us or by others inspired by this work.

(4) Do the authors acquire MS spectra simultaneously with the XPS spectra? MS results would provide additional information about the reactants and products.

Thank you for the suggestion. The MS setup available at the HIPPIE beamline is too slow to register the fast conversion of gas phase products; something we have observed in past experiments. However, intrinsically to APXPS, the C 1s and O 1s gas phase spectra relay the same information as a MS would reveal if it could probe locally with microsecond time resolution. Therefore, we have an internal local gas monitoring system with time resolution on par with the spectral changes we see in surface intermediates. Moreover, the time structure of the obtained gas phase data is identical to the time structure of the surface data due to the hardware synchronization. Therefore, we did not perform MS for this experiment.

We included an explanation of the benefits of using APXPS over MS in the Methods section on p. 11 (l. 440 – 444).

“Note that the APXPS detection of gas phase species is better than conventional mass spectrometry (MS): First of all, it benefits from the synchronization of gas injections and high time resolution of the DLD. Secondly, it probes the local gas composition in the near vicinity of the sample surface (as detailed in Supplementary Fig. 6 of Küst et al.⁹⁶). Therefore, MS data was not recorded in the present study.”

(5) The kinetic energy of emitted electron from O 1s and Pt 4f is set as ~ 200 eV. Why ~500 eV for C 1s?

To simplify the number of changes needed in acquisition parameters, we kept the same photon energy for O 1s and C 1s resulting in a kinetic energy of 500 eV for C 1s. We believe this should not contribute to a difference in our observations for this particular core-level since carbon, in contrast to Pt, are known to be confined to the surface. Another reason is that the beamline delivers better flux at 750 eV and the attenuation in the gas phase becomes lower resulting in faster spectral recording of C 1s surface and gas phase spectra. Importantly, this does not affect our mechanistic conclusions.

We added a sentence to explain the reason for the C 1s photon energy in the Methods section on p. 11 (l. 436-437),

“As adsorbed CO is always located on the surface, we did not tune the photon energy to be surface sensitive for acquiring C 1s spectra.”

(6) It is interesting to observe that during the falling edge, Pt surface oxide is formed while chemisorbed oxygen species are nearly absent. Does this imply that Pt reacts directly with O₂ to form surface oxide rather than through chemisorption followed by reaction?

Although it might be difficult to notice in the figure, the falling edge has a small but detectable amount of chemisorbed oxygen in the presence of a higher surface oxide signal, therefore, it follows that O₂ initially dissociates to form chemisorbed oxygen before reacting in parallel to form surface oxide and react with CO adsorbates to produce CO₂. We detect a large amount of surface oxide signal in comparison to chemisorbed oxygen because the surface oxide is much less reactive than chemisorbed oxygen and thus remains on the surface while chemisorbed oxygen is being rapidly consumed via the two pathways stated above.

We added an inset to Figure 3c to emphasize the presence of chemisorbed oxygen at the start of the falling edge and show the areas proportion between chemisorbed oxygen and surface oxide. We also added text to accompany the inset, “... *the small amount of O_{chem} intensity...*” in the O 1s discussion of the Results section on p. 7 (l. 226).

(7) The authors are encouraged to make the 2D plots and spectra more succinct and clearer to help readers understand the changes in different species.

We agree that the initial figures contain a dense amount of information that has been difficult to present in a clear, straightforward way. To this end, we modified Figures 1 and 2 (now Figures 2 and 3) to help aid the reader in understanding the data. In Figure 1 (now Figure 2), we introduced icons representing each pulsing region (before pulse, rising edge, on pulse, falling edge) with identification of where specific spectral fit examples can be found on the pulse profile (I, II, III, and IV). We incorporated banners with takeaway messages across important changes in the fitted spectra to indicate the importance of the differences observed

on the rising and falling edges compared to on or before pulse regions. In Figure 2 (now Figure 3), we added timeframes with associated time steps to each pulsing region at the top of the figure and physically broke up the pulsing region data to emphasize the nonlinearity of the timescale.

Reviewer #2 (Remarks to the Author):

In this study, Eads et al. investigated CO oxidation on Pt(111) using tr-APXPS to distinguish between active catalytic species and spectator species under millisecond time scales. By modulating CO pulses in an O₂ stream, they identified chemisorbed oxygen as the primary reactive species responsible for CO₂ formation, while the surface oxide primarily acted as a spectator. Although this study confirms the well-established Langmuir-Hinshelwood (LH) mechanism, it significantly enhances our understanding of the structure-activity relationship and enables dynamic control of CO₂ production through CO pulsing. Therefore, while this study is suitable for publication, the following points must be addressed first.

1. These measurements were conducted at 333°C, which is well above the typical temperature range for CO oxidation reactions on Pt surfaces. Are these findings relevant to the species present on the Pt surface—such as the oxide phase, adsorption sites, and the coverage of chemisorbed oxygen and CO—under more realistic reaction temperature conditions?

A reaction temperature of 333 °C was chosen for several reasons: 1) to promote CO oxidation on terraces rather than step edges requires a sufficiently high temperature. This reaction chemistry also mimics a more realistic catalyst operation. 2) Inherent to our gas pulsing experiment, we are limiting the amount of CO gas in the system to bursts of 120 μs. In order to detect CO₂ products within this timescale, it was necessary to run at a higher temperature. Moreover, our reaction conditions allowed us to observe the controversial mixed state of surface oxide and chemisorbed oxygen and determine reactivity under these conditions.

We have a statement about the use of this particular reaction temperature in the Discussion and Conclusions section on p. 9 (l. 301-302),

“This reaction was shown to predominantly occur on the terraces, with the step edges playing a minor role at the temperature used in the study⁶⁵⁻⁶⁷.”

2. In Figure 2a, O₂ GP peak seems to be increasing during rising edge and decreasing during falling edge. The reason of this behavior needs clarification.

Thank you for your keen observation. We believe this is an effect of the close proximity of the O₂ and CO pipes at the front of the Pt sample facilitating the introduction of O₂ into the CO pipe between pulses that is subsequently pushed out when CO is pulsed. However, we’ve noticed that this effect depends on the partial pressures of CO and O₂. In our current study, we observe an increase in the O₂ signal on the CO pulse with a low percentage of CO whereas in past studies (e.g. ref 56) we have observed the opposite effect resulting in a decrease in O₂ signal, effectively blowing the O₂ away, when using a high percentage of CO. Another effect could be associated with attenuation changes upon the introduction of CO where photoelectrons scatter less with CO than O₂ leading to an increase in O₂ signal due to less attenuation.

We included a statement about these effects in the Methods section on p. 11 (l. 410-414):

“Note that the close proximity of the gas pipes at the front of the sample resulted in O₂ gas entering the CO pipe between CO gas pulses and was subsequently pushed out when pulsing as observed in an increase in O₂ gas phase signal (Fig. 2b, Fig. 3a). Gas attenuation may also play a role as photoelectrons will scatter less with CO than O₂ leading to an increase in O₂ signal with the introduction of CO.”

3. Oxide terms and their surface coverage that is used in this study is not well-defined. Reactivity difference with respect to chemisorbed O is clear. In discussion section the authors hypothesize PtO₂ and Pt-O cluster formation, however the type of oxide before the pulse, rising edge and falling edges need better description.

We cannot directly determine the atomic structure of the oxide using our XPS-based technique. However, we can provide convincing evidence for an indirect assignment based upon O 1s and Pt 4f_{7/2} binding energies by comparing with reported values for Pt oxides where they combine both XPS and structure measurements to effectively assign different Pt oxides. The bulk α -PtO₂ tri-layer oxide is quite distinct at higher binding energies (73.5-74.1 eV) and is usually observed at higher O₂ pressure and temperature. Pt surface oxide (sometimes described as Pt-4O) is within the range of 1.1-1.4 eV from Pt bulk and our binding energy assignment fits nicely within this range at 1.4 eV from Pt bulk.

We elaborated upon this topic in the Discussion and Conclusions section on p. 9 (l. 320-325),

“Under the present experimental conditions of gas pressure, temperature, and exposure time, we can infer that we likely have a 4O-type¹⁴ surface oxide (O_{oxide}) that coexists with O_{chem} on metallic Pt, based upon the expansive literature surrounding oxidation of Pt and identification via O 1s and Pt 4f binding energies⁷⁹⁻⁸³. Bulk or surface α -PtO₂ trilayer oxide appear at a much higher binding energy of 73.5-74.1 eV and require higher O₂ pressure and temperature^{59,84-86}.”

4. Is the type of the oxide forming during falling edge the same as the as initial one? It is interesting that oxide appears before the chemisorbed oxygen in falling edge. What would be the mechanism of the formation of this oxide phase in the presence of adsorbed CO and free Pt?

Yes, we believe the surface oxide is the same from the falling edge to the rising edge due to similar kinetic profiles that are mirrored between the two pulsing regions. Although it's difficult to see, the falling edge has a small amount of chemisorbed oxygen in the presence of a higher surface oxide signal. Therefore, it follows that O₂ initially dissociates to form chemisorbed oxygen before reacting in parallel to create surface oxide and react with CO adsorbates to produce CO₂.

We added an inset to Figure 3c to emphasize the presence of chemisorbed oxygen at the start of the falling edge and show the areas proportion between chemisorbed oxygen and surface oxide. We also added text to accompany the inset, “... the small amount of O_{chem} intensity...” in the O 1s discussion of the Results section on p. 7 (l. 226).

5. MvK mechanism involves diffusion of oxide ions from the bulk, perhaps diffusion on the surface could be possible. What is the evidence that removal of the oxide follows MvK mechanism? One can also argue that adsorption rate of CO and its sticking coefficient is different on or near oxide phase. Moreover, the adsorption site of CO in MvK mechanism of the oxide phase is not clear.

The rising edge provides a glimpse into the reaction mechanism and the role of CO as we observe different kinetics in chemisorbed oxygen vs. surface oxide when reacting with CO adsorbates to form CO₂. If CO were reacting with the surface oxide itself then we would expect similar kinetics to chemisorbed oxygen as oxides that have available coordinately unsaturated sites (i.e. PdO(101) and RuO₂(111)) that are highly reactive towards CO. Instead, chemisorbed oxygen reacts faster than surface oxide implying that the surface oxide is missing coordinately unsaturated sites, the oxygen-terminated surface oxide of Rh(111) has a similar structure and they observe inactivity towards the reaction. Therefore, CO instead adsorbs to the surface next to the oxide and reacts with oxygen sites at the edge to slowly eat away the oxide from the outside, amounting to a slower kinetic profile.

In terms of oxide ion diffusion, the surface oxide that we observe likely starts out with a few chemisorbed oxygen atoms surrounding a Pt atom that then forms a PtO₄ cluster which acts as a building block for a 4O-type surface oxide. This sparse, thin surface oxide would likely not contribute to diffusion of oxide ions unlike a proper bulk oxide, such as, the bulk tri-layer α-PtO₂ oxide.

We include a statement about these mechanistic details in the Discussion and Conclusions section on p. 9 (l. 330-334),

“Furthermore, the diminished Pt_{surf} intensity with the stabilization of O_{oxide} may contribute to the lack of reactivity based upon the observed inactivity on a similar oxygen-terminated surface oxide on Rh(111) due to missing coordinately unsaturated sites⁸⁷ as compared to highly reactive PdO(101)⁸⁸ and RuO₂(110)⁸⁹ with available coordinately unsaturated sites.”

6. Ptsurf remaining until 1.2 ms (Figure 2d) despite surface already dominated by CO is not realistic. Could this be a fitting error? The authors must comment on this.

We believe our fits are justified and that the Pt_{surf} tail is not simply a fitting error but rather physically meaningful up to 0.8 ms. We first confirmed the need for Pt_{surf} in the fit by overlapping the raw spectra (background-corrected), normalized to the Pt bulk feature, to visualize the low binding energy tail where Pt_{surf} is located. Indeed, we observe enhanced intensity in the low binding energy region compared to the initial surface before the CO pulse, up until the fit no longer incorporates Pt_{surf}. Secondly, CO_{bridge} signal in Fig. 3b increases over 0.4-0.8 ms with the simultaneous decrease of Pt_{surf} signal in Fig. 3d. Suggesting that CO continues to adsorb on available Pt surface atoms in a bridged configuration during this time. Beyond this time from 0.8-1.2 ms, the error bars associated with the fit become significant compared to the value of the Pt_{surf} area and could be considered within the noise of the fit.

To clarify this point, we added the following statement to the Results section on p. 7 (l. 238-243),

“At CO_{ads} saturation, Pt_{surf} signal declines exponentially, potentially a result of slow Pt atom diffusion originating from Pt atoms/clusters that are left behind when CO_{ads} reacts with oxygen atoms in O_{oxide}. Moreover, CO_{bridge} signal increases as Pt_{surf} signal decreases from 0.4-0.8 ms suggesting that CO molecules are preferentially adsorbing in a bridged configuration as Pt sites become available. Pt_{surf} signal continues to decline until ~1.2 ms but can be considered within the noise of the fit from 0.8-1.2 ms.”

7. Minor comment: On page 3, the authors report that they monitor surface and gas phase evolution below 1 ns time resolution. I believe this is a typo. If this has something to do with

DLD counting, I think it requires rephrasing since the time resolution of this work is not in ns range.

Thank you for the observation. It is correct that this particular time resolution is attached to DLD counting rather than the time resolution associated with our experiment. We have since removed "(below 1 ns)" in the Introduction section to avoid confusion and added a comment in the Discussion and Conclusions section to show the potential of an even better time resolution which is of course currently dependent upon the gas pulsing valves.

"Even better time resolution could be possible in the future with a faster piezo valve to match the nanosecond time resolution of the DLD." (p. 10, l. 381-382).

8. Minor comment: The color or position of CO_{top} and CO_{bridge} in Figure 1c should be adjusted to avoid confusion with chemisorbed O during the falling edge.

Alongside other reviewer comments, we have modified Figures 1 and 2 (now Figures 2 and 3) to help aid the reader in processing the dense information provided in both figures. To your comment, we made all the text in the 2D plots black and moved down the labels for CO_{top} and CO_{bridge} to align with the area in which CO signal is most prominent.

Reviewer #3 (Remarks to the Author):

This is an important work, on one hand reporting the implementation for the first time of efficient pump-and-probe Ambient Pressure X-Ray photoelectron Spectroscopy (APXPS) using periodical chemical perturbations -gas pulses- to achieve sub-millisecond time resolution, and on the other hand using this innovation to cast light on the detailed mechanisms of the thoroughly investigated CO oxidation reaction on Pt(111), with potential impact also on the optimization of dynamic catalyst operation.

Compared to a previous publication from the same group, ref. 56, corresponding to a preceding stage of their time resolved APXPS project, now the method achieves its full potential with the upgrading of the the camera to a DLD detector, which enables acquiring all the time delays for each kinetic energy in a single gas pulse, increasing the efficiency by one or two orders of magnitude.

The method enabled the detection of O_{chem} and O_{oxide} species, and their time correlation with high and intermediate CO oxidation activity, respectively, making it possible to deduce the coexistence of the two proposed reaction mechanisms, LH and MvK. This is relevant not only from a fundamental perspective, as it allows for nicely harmonizing the broad amount of information on this reaction, but it also has great significance in the understanding and optimization of industrial CO oxidation using pulsed gas.

In summary, this is an outstanding work, and I recommend its publication in Nature Communications after consideration of the following minor comments:

a) It would be useful to detail how many CO pulses and total acquisition time were needed for the data in Fig. 1.

Each 2D plot in Figure 1 required 2.5 hours averaging over 90,000 CO pulses when pulsing at 10 Hz. Considering all 2D plots in Figure 1, the total acquisition time was 12.5 hours averaging over 450,000 CO pulses when pulsing at 10 Hz.

We added a comment about the acquisition time in the new gas pulsing methodology subsection at the start of the Results section on p. 3 (l. 134-136),
“Pertinent to our current study monitoring CO oxidation over Pt(111), Fig. 1b presents a 2D plot of C 1s gas phase when pulsing CO gas at 10 Hz into an O₂ stream averaged over 90,000 pulses over 2.5 hours.”

b) At first sight, the non-linear time scale of Fig. 2, makes one -or me- believe that the surface was most of the time in CO-poisoned state. This additionally leads to a difficulty understanding why the CO pulsing frequency was in a later experiment increased to 50 Hz (if the surface was already poisoned at 10Hz, I thought). While it is true that in Fig. 1 it is clearly seen that it is not the case that at 10 Hz the surface is most of the time poisoned, it might still be useful to call attention in the text or in the caption to the non-linear time scale in Fig. 2.

We have also noticed this and struggled to find a way to present the nonlinearity of the timescale in a way that wasn't overwhelming to view. We hope we have managed to do this now by adding a label on top of Figure 2 (now Figure 3) with information about the duration of each pulsing region with their associated timesteps and physically dividing each pulsing region with breaks in between.

c) In the pulse vs flow experiments reported in the Supplementary Note 1, was the temperature also T=333C?

Yes, the temperature remained at 333 °C throughout our presented studies.

We have added this detail in the Methods section on p. 12 (l. 473-475),
“CO pulsing vs. flow experiments required adjusting either the CO mass flow controller (4 sccm) or CO pulsing voltage (170-240 V), respectively, to achieve the targeted CO:O₂ ratios, while O₂ remained at 30 sccm output throughout the experiments and the system heated to 333 °C”

d) In line 105: “.. we revolutionized”, I think that it should read: “we revolutionize”

Thank you for noticing this, we have corrected “revolutionized” to “revolutionize”.

e) In line 107: monitor the surface and gas phase evolution with high (below 1 ns)...”, it should be “... (below 1 ms)..”

We realize now that this is confusing based upon your comment and another reviewer's comment. This particular time resolution is associated with DLD counting. We have since removed this statement from the Introduction section to avoid confusion and added a comment in the Discussion and Conclusions section to show the potential of an even better time resolution which is of course currently dependent upon the gas pulsing valves.

“Even better time resolution could be possible in the future with a faster piezo valve to match the nanosecond time resolution of the DLD.” (p. 10, l. 381-382).

f) In line 184: would it be possible to provide an estimate of the residence time of CO in the Pt(111) surface, rather than defining it as “extremely low” or “CO reacts instantaneously”? Is

it of the order of ns?

Based on the literature data on the measured desorption energy and pre-exponential factors (ref. 57), we estimate the residence time of CO molecules on the terraces of Pt(111) to a few hundred microseconds for thermal CO desorption. However, our system is different due to the reaction of CO with oxygen adsorbates, which is likely faster than thermal CO desorption. Therefore, we expect that CO residence time is less than a few hundred microseconds.

We corrected the information in the Results section on p. 7 (l. 213),
"...visible due to the low residence time of CO on the surface (likely less than thermal CO desorption at a few hundred μs^{57}) ..."

g) In line 226 it is stated that the time step for fitting the O1s and C1s 50 Hz data was 1ms/50 spectra, however in the "Methods" section it is said that it was 280us /7 spectra.

Thank you for noticing this discrepancy. The information in the Methods section includes the fitting information from our first iteration of processing using our normal procedure (same as 10 Hz), however the low signal-to-noise ratio for the 50 Hz dataset required us to average more spectra in order to fit it properly.

We corrected the information in the Methods section on p. 12 (l. 469-470),
"Note that the 50 Hz dataset required averaging over 1 ms / 50 spectra to achieve a reasonable signal-to-noise ratio for fitting."